# Vaccination Status, Vaccine Awareness and Attitudes, and Infection Control Behaviors of Japanese College Students: A Comparison of 2021 and 2023

**DOI:** 10.3390/vaccines12090987

**Published:** 2024-08-29

**Authors:** Yuri Okamoto, Takahito Yoshida, Tatsuhiro Nagata, Yui Yumiya, Toru Hiyama, Yoshie Miyake, Atsuo Yoshino, Shunsuke Miyauchi, Tatsuhiko Kubo

**Affiliations:** 1Health Service Center, Hiroshima University, Higashihirsohima 739-0046, Japan; tohiyama@hiroshima-u.ac.jp (T.H.); miyakechare@hiroshima-u.ac.jp (Y.M.); yoshino@hiroshima-u.ac.jp (A.Y.); smiyauchi@hiroshima-u.ac.jp (S.M.); 2Department of Public Health and Health Policy, Graduate School of Biomedical and Health Sciences, Hiroshima University, Hiroshima 734-8553, Japan; b173309@hiroshima-u.ac.jp (T.Y.); d222525@hiroshima-u.ac.jp (T.N.); yumiya@hiroshima-u.ac.jp (Y.Y.); tkubo@hiroshima-u.ac.jp (T.K.)

**Keywords:** vaccination, after COVID-19, infection control, college students

## Abstract

**Background**: Now that the spread of COVID-19 has been controlled, it is important to investigate changes in young people’s perceptions of the vaccine and their behavior toward infection. The objectives of this study were as follows: (1) to investigate the association between Omicron strain vaccination rates among college students, their perceptions of the vaccine, and past adverse reactions to the vaccine; (2) to compare 2021 (when COVID-19 was spreading) and 2023 (when COVID-19 was strained) to identify changes in attitudes toward vaccination and motivations for vaccination and changes in infection prevention behavior. **Methods**: This cross-sectional survey was conducted via e-mail from 5 January to 30 January 2023. All students at Hiroshima University were sent an e-mail, which provided them access to the survey form and requested their cooperation. The questionnaire consisted of 33 items related to attributes, vaccination status, adverse reactions after vaccination, motivation for vaccination, perception of the vaccine, presence of coronavirus infection, sequelae, and infection prevention measures. **Results**: A total of 1083 students responded to the survey. Over 50% of the students were vaccinated with the Omicron booster. Regarding trust in vaccines, the majority of both male and female respondents said they had some trust in vaccines, although this was less than that observed in the 2021 survey. As for infection control measures, only 2% of males and 0.3% of females answered that they did not take any infection control measures. The most common response was “wear a mask”, as in the 2021 survey, with 476 men (96.6%) and 575 women (99.5%). **Conclusions**: The survey showed a high Omicron-responsive vaccination rate of more than 50%. In addition, more than 99% of the students were found to be taking measures to prevent infection, such as wearing masks.

## 1. Introduction

The WHO Emergency Committee raised international concerns on 23 January 2020 [1], and characterized the outbreak of COVID-19 as a pandemic [2]. The Japanese government instructed its citizens to take thorough precautions to prevent infection, including wearing masks, washing hands, avoiding crowded places, and promoting ventilation [3,4]. Vaccination was also recommended as a preventive measure. However, there have been reports of concerns about the public’s awareness of vaccines and the acceptability and intent of vaccination [5,6], especially in young adults [5,7,8,9].

In Japan, free vaccination began in 2021. To promote vaccination among young people, the Japanese government recommended mass vaccination programs at universities. In response, the Health Service Center at Hiroshima University initiated a large-scale vaccination program. Accurate information about the vaccine was disclosed on the university’s websites, and vaccine reservation forms were sent to all constituents. In addition, information on adverse reactions to the vaccine was provided at the time of vaccination and contact information (Health Service Center) was clearly provided in cases of adverse reactions. In addition, we believe that infection prevention measures, such as wearing masks, are important even after vaccination, and we informed college students of the need for infection prevention measures after vaccination. We conducted a cross-sectional survey via e-mail from September to October 2021 on vaccination coverage, attitudes toward vaccines, and the post-vaccination behavior of students and faculty members to ascertain the actual situation among young people. The vaccination rate was high at 76.5% for students and 86.7% for faculty and staff, and, while many students indicated that their range of activities had expanded, many students indicated that they had not changed after vaccination with regard to infection prevention measures (87.8%). The fact that young people who engage in a wide range of activities pay attention to infection prevention is considered to be one of the factors preventing the rapid spread of infection.

The Director-General of the WHO determined, on 5 May 2023, that COVID-19 is now an established and ongoing health problem that no longer constitutes a public health emergency of international concern (PHEIC) [9]. As of 8 May 2023, the Japanese Ministry of Health, Labor and Welfare (MHLW) moved COVID-19 from category 2 (e.g., tuberculosis and SARS) to category 5 (e.g., seasonal influenza) of the classification (categories 1–5) based on the Infectious Disease Law [10]. The government no longer uniformly requires people to refrain from leaving the house or to take measures against infection, leaving it up to individuals to make their own decisions. Now that college students are returning to their daily campus life, it is important to investigate what changes are occurring regarding young people’s perception of vaccines and their behavior toward infection prevention.

The objectives of this study were as follows: (1) to investigate the association between Omicron strain vaccination rates among college students, their perceptions of the vaccine, and past adverse reactions to the vaccine; (2) to compare 2021 (when COVID-19 was spreading) and 2023 (when COVID-19 was contained) to identify changes in attitudes toward vaccination and motivations for vaccination and changes in infection prevention behavior.

## 2. Materials and Methods

### 2.1. Participants

This cross-sectional survey was conducted via e-mail from 5 January to 30 January 2023. All students (10,603 people) at Hiroshima University were sent an e-mail, which provided them access to the survey form and requested their cooperation in the survey.

This survey system was conducted using the same methodology used in our 2021 survey [8], and was designed to allow respondents to access and respond to the survey instrument through a personal e-mail request. Each respondent could only respond once, thus preventing duplicate responses.

### 2.2. Ethical Considerations

This study was conducted in accordance with the guidelines proposed in the Declaration of Helsinki and was approved by the Epidemiology Ethics Committee of Hiroshima University (approval ID: E2022-0302, 10 May 2023). When we sent the survey by personal e-mail and requested the cooperation of the student, we told them that their responses would be handled anonymously and that they would not be disadvantaged if they did not cooperate with the survey. It was assumed that consent was obtained by cooperating in the survey.

### 2.3. Measures

The questionnaire consisted of 33 items related to attributes, vaccination status, adverse reactions after vaccination, motivation for vaccination, perception of the vaccine, presence of coronavirus infection, sequelae, and infection prevention measures. Questions about the participants’ psychological situation included the presence or absence of fear of the vaccine and their impression of vaccination (multiple-choice answer). Students who were vaccinated for the Omicron variant were asked about their motivation for receiving the vaccine (multiple-choice answer). The participants were also asked about preventive actions they were currently taking after the pandemic (multiple-choice answer). The questionnaire was constructed by collecting the opinions of internists, psychiatrists, public health experts, university administrators, and faculty members. The participants were determined to “trust” the effectiveness of the vaccine if they answered “fully” or “somewhat trust” in the effectiveness of the vaccine.

A similar survey was conducted with students in 2021 [8]. The results of that survey and the 2023 survey were compared in terms of vaccination status, concerns about vaccination, trust in vaccination, and motivations of vaccinators. The extent to which students are taking infection prevention measures was also investigated.

The results of the 2021 survey (2160 respondents) and the 2023 survey (1083 respondents) were compared. Responses to the survey were voluntary and were small, just over 10%.

### 2.4. Statistical Analysis

We used a chi-square test for comparison of the two groups with and without trust in vaccines, the two groups with and without anxiety about vaccines, and between 2021 and 2023. Statistical analysis was performed using JMP Pro for Mac version 16 (SAS Institute Japan, Tokyo, Japan). A two-sided *p*-value less than 0.05 was considered to indicate statistical significance.

## 3. Results

### 3.1. History of COVID-19 Infection and Sequelae Symptoms

A total of 1083 (493 males and 590 females) students responded to the survey. COVID-19 affected 139 males and 168 females. Among the affected students, 73 (52.5%) males and 86 (51.2%) females reported having sequelae symptoms. There were no significant differences between males and females in terms of either morbidity or the incidence of symptoms. Cough and shortness of breath were the most frequent symptoms (males 33.8%, females 33.3%), followed by fatigue (males 22.3%, females 25.6%), taste and smell loss (males 17.3%, females 14.9%), brain fog (males 7.2%, females 4.8%), joint pain and muscle aches (males 6.5%, females 4.8%), and others (males 5.0%, females 5.4%). There were no significant differences between males and females. No patients complained of serious symptoms.

### 3.2. Vaccination Status of Omicron-Responsive Vaccines, Perception of Vaccines, and Past Adverse Reactions

Table 1 shows the results for participants with and without Omicron booster vaccination. Over 50% (males 52.9%, females 53.5%) of the students were vaccinated with the Omicron booster. Among those who received the booster vaccination, 243 (93.1%) males and 277 (89.6%) females had an adverse reaction to the previous vaccination. Among those who had not received a booster vaccination, 188 (81.0%) males and 235 (87.4%) females had an adverse reaction to the previous vaccination. There were no significant differences in the presence or absence of adverse reactions between males and females.

As for the participants’ perception of the vaccines, the most common response was that they had some trust in the vaccines, with a majority of males (258, 52.3%) and females (343, 58.1%) having some trust. This group was followed by 186 (37.7%) males and 253 (42.9%) females who had anxiety about adverse reactions. Furthermore, 133 (27.0%) males and 94 (15.9%) females responded that they expected it, and 33 (6.7%) males and 52 (8.8%) females responded that they were afraid of it.

Regarding previous presence or absence of adverse reactions and trust in vaccines, those who had no adverse reactions were significantly more likely to trust the vaccines (*χ*^2^(1) = 3.794, *p* < 0.01, Phi = 0.006). Regarding previous presence or absence of adverse reactions and anxiety about the vaccines, those who experienced adverse reactions were significantly more likely to experience anxiety about the vaccines (*χ*^2^(2) = 4.961, *p* < 0.01, Phi = 0.007).

Figure 1 shows the motivation for vaccination with the Omicron-compatible vaccine, based on the responses of 276 on-campus vaccination recipients (in response to a question to those who received the mass intake at Hiroshima University). The most common answer was “to prevent serious illness”, followed by “because there was a call for vaccination” and “because I want to contribute to society”.

### 3.3. Implementation of Individual Infection Control Measures

Responses regarding the infection control measures that were still being taken by participants are shown in Figure 2. The largest number of respondents (476 (96.6%) males and 575 (99.5%) females) indicated that they would wear a mask. Proactive hand washing was next, with 370 (75.1%) males and 482 (83.4%) females. Ten males (2.0%) and two females (0.3%) reported taking no special infection control measures.

### 3.4. Comparison with the 2021 Survey

Table 2 shows the comparison between 2021 and 2023. Compared to the survey conducted in October 2021 (during the fifth wave of the virus spread in Japan), 85 of 1083 (7.9%) respondents reported fear of the vaccine in this survey, which was significantly less than that reported in the 2021 survey (842/2050 (41.1%), *χ*^2^(1) = 434.597, *p* < 0.0001). A total of 601 of 1083 (56.1%) respondents in the current survey reported having trust in the vaccines, compared to 1828/2050 (89.2%) in the 2021 survey (*χ*^2^(1) = 444.244, *p* < 0.0001). Regarding the motivation for vaccination, 166/276 (60.1%) respondents in the current survey reported that they were vaccinated to contribute to the prevention of infection (society, elderly, family), which was less than the 1378/2050 (67.2%) respondents with the same answer in the 2021 survey (*χ*^2^(1) = 8.192, *p* = 0.004).

## 4. Discussion

### 4.1. Vaccination Rates and Promotion

The spread of COVID-19 infection and the severity of its symptoms were partly due to hesitancy to vaccinate [11], in addition to a lack of preventive behavior [12]. A large amount of COVID-19 disinformation on social media, including conspiracy theories about COVID-19, has been reported to potentially influence many people’s opinions and beliefs about COVID-19 [13,14,15]. Several reports [16,17,18,19,20] have suggested that a higher level of understanding and awareness of COVID-19 among students, in terms of preventive measures, can reduce the spread of the disease.

In our survey of Japanese college students during the COVID-19 infection spread period in 2021 [8], we found that 76.5% of college students had received two doses of the vaccine in a large-scale university-wide vaccination program, and that the wide range of support, including information dissemination about the vaccine, COVID-19, and mental care for anxiety, was effective. Although the vaccination rate in the current survey was lower than that in the previous survey, the vaccination rate among students who received the Omicron booster was more than 50%, suggesting that information dissemination and awareness-raising among college students may have contributed to higher vaccination rates.

Furthermore, in the present study, conducted under conditions in which the spread of infection had subsided, we found that more males and females who had been vaccinated with a booster had adverse reactions to vaccination, compared to those who had not been vaccinated with boosters. The results suggest that the presence of adverse reactions was not a reason for hesitating to take the booster. The majority of both males and females reported that they trusted the vaccine. It is thought that the respondents may have gained confidence in vaccination as their understanding and knowledge of the adverse reactions and effects of vaccines have improved. 

### 4.2. Infection Control Measures

In this survey, most participants took infection control measures such as wearing masks and washing their hands. The importance of infection control measures such as wearing masks and washing hands has been reported [21,22], especially that wearing masks is also important in preventing other infectious diseases [23,24]. This may indicate that the importance of infection control measures has become well known due to the spread of COVID-19 infection. Several studies have found that higher levels of education are associated with higher levels of compliance with hand-washing and mask-wearing policies [25,26,27,28]. Providing information about the importance of continuing infection prevention measures, even after the spread of infection had ceased, seemed to have influenced the behavior of the college students.

Kimbler et al. argue that normative beliefs are important, and report that normative beliefs predict preventive behavior, vaccine planning, and vaccination status [15,29,30,31]. The fact that many students in this survey continued to take infection prevention measures, such as wearing masks and washing their hands, even after COVID-19 was removed from the list of public health emergencies of international concern may be related to more accurate information and understanding, as well as a stronger sense of normality among Japanese students.

Several reports discuss preventive behavior in relation to health literacy. Zhang et al. [32,33,34,35,36,37] stated that vaccine literacy (VL) is an important component of health literacy, and further stated that public health education should actively support the development of public health education with content appropriate to the characteristics and preferences of youth. In this line, we disseminated information via e-mail and the internet; however, it is also important for the spread of education to incorporate social networking services and other methods of information dissemination tailored to the preferences of young people.

### 4.3. What We Can See from Comparison with the 2021 Survey

The decrease in vaccination coverage in 2023 may have been due in part to vaccination fatigue among students. Compared to the 2021 survey, vaccination fatigue among students may be a contributing factor to the lower immunization rate. Fear of the vaccine decreased significantly, but trust in the vaccine also decreased significantly. In this study, providing accurate information, including negative information about adverse reactions to vaccines, was effective and may have significantly reduced fear of vaccines. The high trust in 2021 reflects the increased expectations for vaccines due to the threat of the spread of infection. The decrease in trust seemed to be influenced by the fact that the crisis eased after the spread of the infection subsided. The large decrease in fear of vaccines may have been because of providing accurate information, including negative information about adverse reactions to vaccines.

Furthermore, both males and females have significantly less trust in the vaccine than in 2021. However, more than 50% still trust the vaccine, suggesting that although blind trust declined due to the crisis during the 2021 epidemic, the correct attitude may be adopted now that there is greater knowledge about COVID-19, vaccine efficacy, and adverse reactions.

Motivation for vaccination “to contribute to infection prevention” also decreased significantly from 2021, but more than 60% of respondents answered, “to contribute to infection prevention”. Lee et al. [38,39] discussed the effectiveness of awareness-raising and education among young people regarding preventive behavior. Awareness-raising and the education of young people may be a major key to future infectious disease control. Lee et al. [38,39] argue that educating young people about preventive behavior is effective. In the present study, it is possible that awareness-raising and education were effective for college students. 

### 4.4. Advantage and Limitation of This Study

The advantages of this study are that the questionnaires were previously administered in the same setting as the mass vaccination program at the university in 2021, the 2023 survey was as feasible as the 2021 survey, and the data could be collected quickly.

One of the limitations of this study is the number of respondents. The 2021 survey (2160 respondents) and the 2023 survey (1083 respondents) also had small response rates (10–20%). The response to the survey was voluntary, and it is possible that only those who were interested responded to the survey. Furthermore, the number of respondents decreased this time compared to 2021. This may be related to somewhat less interest in taking the vaccine compared to 2021, when the infection is spreading globally. In addition, it was not possible to compare the same subjects in the 2021 and 2023 surveys. Furthermore, because the questions were not the same, it was difficult to capture changes in detail.

In preparation for a future pandemic caused by a new type of infectious disease such as new coronas, we would like to understand young people’s understanding and awareness of infectious diseases through a large-scale survey and explore the potential for change and sustainability through educational interventions.

## 5. Conclusions

The survey revealed that the Omicron-responsive vaccination rate among Hiroshima University students is high, at more than 50%. 

Compared to our survey in 2021, both fear and trust in vaccines decreased significantly. However, the majority of males and females reported that they trust the vaccine, suggesting that although blind trust decreased due to the crisis during the 2021 epidemic, the correct attitude may be adopted now that there is greater knowledge about COVID-19, vaccine efficacy, and adverse reactions.

The study also found that more than 99% of students use masks and other infection-prevention measures. Raising awareness and educating the younger generation about the correct knowledge of infectious diseases, vaccines, and preventive actions may help society as a whole take action against infectious diseases.

## Figures and Tables

**Figure 1 vaccines-12-00987-f001:**
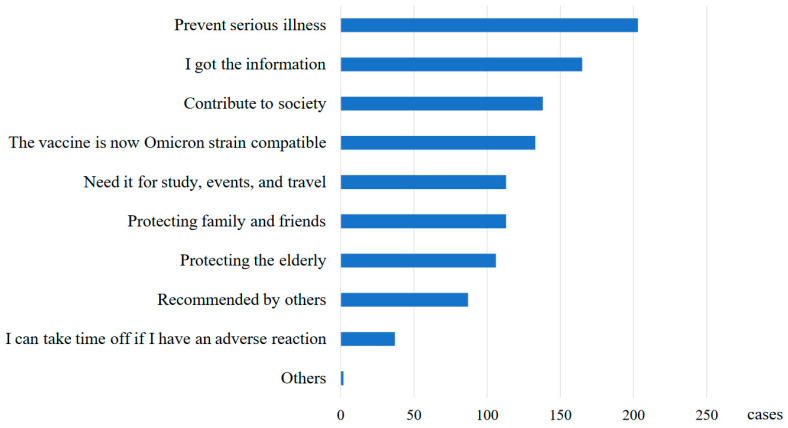
Motivation for Omicron-specific COVID-19 booster vaccination (multiple answers).

**Figure 2 vaccines-12-00987-f002:**
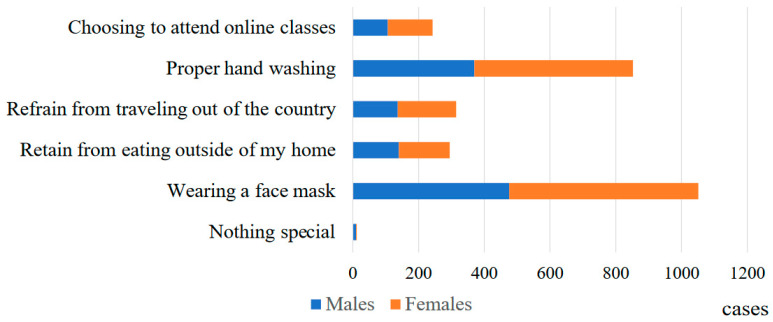
Infection prevention measures (multiple answers).

**Table 1 vaccines-12-00987-t001:** Omicron-specific COVID-19 booster vaccination rates and incidence of adverse reactions.

Gender	Booster Vaccination		
	Vaccinated	Unvaccinated	Total
Male	261 (52.9%)	232 (47.1%)	493
adverse reaction(+)	243/261 (93.1%)	188/232 (81.0%)	431/493 (87.4%)
Female	309 (53.5%)	269 (93.5%)	578
adverse reaction(+)	277/309 (89.6%)	235/269 (87.4%)	512/578 (88.6%)
Neither	6 (50.0%)	6 (50.0%)	12
adverse reaction(+)	5 (83.3%)	5 (83.3%)	10/12 (83.3%)
Total	574	509	1083

**Table 2 vaccines-12-00987-t002:** Comparison of fear and trust in vaccines and motivation to vaccinate in 2021 and 2023.

	Fear of the Vaccines	Trust in the Vaccines	Motivation for Vaccination Contribute to the Prevention of Infection
2021	842/2050 (41.1%) ** *χ*^2^(1) = 434.597, *p* < 0.0001	1828/2050 (89.2%) ** *χ*^2^(1) = 444.244, *p* < 0.0001	1378/2050 (67.2%) ** *χ*^2^ (1) = 8.192, *p* = 0.004
2023	85/1083 (7.9%)	601/1083 (56.1%)	166/276 (60.1%)

** We used a chi-square test using JMP Pro for Mac version 16 (SAS Institute Japan, Tokyo, Japan).

## Data Availability

The data can be shared up on request.

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
