# Peer review of "Vaccination Status, Vaccine Awareness and Attitudes, and Infection Control Behaviors of Japanese College Students: A Comparison of 2021 and 2023"

_vaccines, 2024, doi:10.3390/vaccines12090987_

Round 1

Reviewer 1 Report (New Reviewer)

Comments and Suggestions for Authors

This study investigated the association between college students' vaccination coverage of the Omicron strain and their image of the vaccine and past adverse reactions to the vaccine and compared the results with those in 2021 to determine any changes. The comments are as follows:

1. In the data collection, 10,603 students were sent email for survey. However, only 1083 students were included in the data analyses. Thus, the representativeness of the participants should be explained. 

2. In figures 1 and 2, the vertical and horizontal axis should be changed with each other. 

3. Recommendation for future research shpuld be addressed.

4. The manuscript needs extensive language correction. 

Comments on the Quality of English Language

The manuscript needs extensive language correction. 

Author Response

Dear Reviewer,

Thank you for your warm comments. We have taken them to heart. It will be an encouragement to us in the future.

Point 1: In the data collection, 10,603 students were sent email for survey. However, only 1083 students were included in the data analyses. Thus, the representativeness of the participants should be explained. 

Response 1: Indeed, as you pointed out, there is a problem with the small number of respondents to the survey. Responses are voluntary and are not representative of the entire population. In this regard, we have noted the “voluntary” nature of the responses in the methodology (Line 85-86), and have also noted this as a limitation of the study (Line 253-257).

Point 2: In figures 1 and 2, the vertical and horizontal axis should be changed with each other.

Response 2: Following your suggestion, in Figures 1 and 2, the vertical and horizontal axes have been interchanged.

Point 3: Recommendation for future research should be addressed.

Response 3: Thank you for your valuable suggestions. We have included them at the end of Discussion (Line 265-268).

Point 4: The manuscript needs extensive language correction.

Response 4: Thank you for your advice. We have considered the suggestions of the other reviewers and revised the manuscript.

Reviewer 2 Report (New Reviewer)

Comments and Suggestions for Authors

Manuscript: Japanese University Students are Highly Conscious of Infection Prevention Even After the Spread of COVID-19

I would like to thank the authors and the Editors for allowing me to evaluate their work. The authors reported the prevalence of Omicron booster vaccination according to gender and previous adverse reaction status. They also compared two surveys on COVID-19, 2021 and 2023, in terms of fear, trust, and motivation for vaccination. Although the topic is important, the report, in its current situation, is not suitable for publication.  

My concerns are the following.

1.       Title:

It does not reflect the study objectives and better be modified.

2.       Abstract:

It needs to be improved, particularly the background section, which needs to be summarized and the methods, which lack important information.   

3.        Introduction:

This section is well written. However, the objectives were not specific and needed attention. The authors mentioned the study objectives as:1) to investigate the association between college students' vaccination coverage of the Omicron strain and their image of the vaccine and past adverse reactions to the vaccine and 2) to compare the results with those in 2021 (when the infection spread) to determine any changes.

No specific statistical association was provided; the authors only reported the prevalence of Omicron booster vaccination according to gender and previous adverse reactions. Further, they compared two surveys on COVID-19, 2021 and 2023, in terms of fear, trust, and motivation for vaccination. I suggest they rewrite the objectives to be more specific.

4.       Methods:

 The authors used the 2021 survey data for comparisons but without any information about this survey in the methods section. Please add information or cite any reference for it may be, your published work  doi: 10.3390/vaccines10060863.

5.     Results:

I strongly recommend modifying this section and specifically the following

-            The Table titles are too general and do not accurately represent the presented data.

-            The Figures are presented as frequencies rather than percentages, making it challenging to draw conclusions.

-            Please provide the actual p values for the three columns in Table 2 and state the statistical test used under the Table.

6.       Discussion  

The discussion is quite general in some sections and lacks clear linking with the main outcome. There is room for improvement, especially in the following issues:

1.       The results should not be repeated in the discussion section. They should be presented more concisely, emphasizing the comparison with previous studies.  

2.       Authors reported very general information in sections that were not directly linked with their results, such as lines 169-174 and lines 212-218.

3.       For Line 189, “ The majority of both males and females reported that they trusted the vaccine to some extent” Data is not provided for Trust in vaccination (totally, to some extent, etc.

4.       Line 185 “ In the present”, in the present.

7.       Conclusion

What are the public health policies or implications of the study's findings?

Author Response

Dear Reviewer,

Thank you for your warm comments. We have taken them to heart. It will be an encouragement to us in the future.

Point 1: Title: It does not reflect the study objectives and better be modified.

Response 1: Thank you for your suggestion. We have changed the title to “Vaccination status, vaccine awareness and attitudes, and infection control behaviors of Japanese college students: A comparison of 2021 and 2023”.

Point 2: Abstract: It needs to be improved, particularly the background section, which needs to be summarized and the methods, which lack important information.

Response 2: We have summarized the background and increased the description of the methods, as you suggested.

Point 3: Introduction: This section is well written. However, the objectives were not specific and needed attention. The authors mentioned the study objectives as:1) to investigate the association between college students' vaccination coverage of the Omicron strain and their image of the vaccine and past adverse reactions to the vaccine and 2) to compare the results with those in 2021 (when the infection spread) to determine any changes. No specific statistical association was provided; the authors only reported the prevalence of Omicron booster vaccination according to gender and previous adverse reactions. Further, they compared two surveys on COVID-19, 2021 and 2023, in terms of fear, trust, and motivation for vaccination. I suggest they rewrite the objectives to be more specific.

Response 3: Thank you for your valuable suggestions. We rewrote the objectives of our study to be more specific (Line 44-50). Statistical analysis is described in the Methods section as 2.4 Statistical Analysis (Line 92-97).

Point 4: Methods: The authors used the 2021 survey data for comparisons but without any information about this survey in the methods section. Please add information or cite any reference for it may be, your published work  doi: 10.3390/vaccines10060863.

Response 4: Following your suggestion, we have noted that we conducted our survey in 2021 and that we compared the results of that survey with the results of the current survey. In addition, we cited our published literature.

Point 5: Results: I strongly recommend modifying this section and specifically the following

  1. The Table titles are too general and do not accurately represent the presented data.
  2. The Figures are presented as frequencies rather than percentages, making it challenging to draw conclusions.
  3. Please provide the actual p values for the three columns in Table 2 and state the statistical test used under the Table.

Response 5: Thank you for your valuable comments.

  1. The title of the table has been corrected. 2. We have changed the figure to show percentages. 3. We have provided the actual p values for the three columns in Table 2 and state the statistical test used under the Table.

Point 6: Discussion: The discussion is quite general in some sections and lacks clear linking with the main outcome. There is room for improvement, especially in the following issues: 1.The results should not be repeated in the discussion section. They should be presented more concisely, emphasizing the comparison with previous studies.  2. Authors reported very general information in sections that were not directly linked with their results, such as lines 169-174 and lines 212-218. 3. For Line 189, “ The majority of both males and females reported that they trusted the vaccine to some extent” Data is not provided for Trust in vaccination (totally, to some extent, etc. 4. Line 185 “ In the present”, in the present.

Response 6: Thank you for your valuable comments.

  1. The results were not repeated in the discussion, but were made more concise. 2. General statements unrelated to the results were removed. 3. We had already described in the Methods section that if participants indicated that they “fully” or “somewhat trust” the effectiveness of the vaccine, then they “trust” the effectiveness of the vaccine. Therefore, in the Discussion, we only stated “trust”. 4.We corrected “In the present” to “in the present”.

Point 7: Conclusion: What are the public health policies or implications of the study's findings?

Response 7: Very valuable point. It is a difficult point, but based on this research, I believe that awareness and education of the younger generation will contribute to infection prevention measures in the future, and that is why I have included it (Line 286-289).

Reviewer 3 Report (New Reviewer)

Comments and Suggestions for Authors

This article generally employs a relatively standardized research method, but there are some issues in the text that need to be revised:

  1. There are some colloquial expressions in the text, such as "Now that college students are returning to their daily campus life, we felt it was important to investigate" in the abstract.
  2. It is unclear whether this article has been revised, as there are highlighted and modified parts in the text.
  3. The conclusion section should provide a summary of findings, which is currently too simplistic.
  4. There are some misuses of punctuation in the text.
  5. The capitalization in the abstract is somewhat inconsistent.

Author Response

Dear Reviewer,

Thank you for your warm comments. We have taken them to heart. It will be an encouragement to us in the future.

Point 1: There are some colloquial expressions in the text, such as "Now that college students are returning to their daily campus life, we felt it was important to investigate" in the abstract.

Response 1: Thank you for your suggestion. In the Abstract and Introduction sections, we have corrected colloquialisms such as “As college students are returning to their daily campus life, we felt it was important to conduct a survey.

Point 2: Abstract: It is unclear whether this article has been revised, as there are highlighted and modified parts in the text.

Response 2: The highlights and corrections were in an earlier manuscript. Our apologies. In this issue, newly revised sections are shown in red.

Point 3: The conclusion section should provide a summary of findings, which is currently too simplistic.

Response 3: Following your suggestion, we have included our results in the Conclusions section and also added a look ahead to the future (Line 279-289).

Point 4: There are some misuses of punctuation in the text.

Response 4: Thank you for pointing this out. We have asked the  editor to correct the English text again and have made the correction.

Point 5: The capitalization in the abstract is somewhat inconsistent.

Response 5: Thank you for pointing this out. We have asked the  editor to correct the English text again and have made the correction.

Round 2

Reviewer 1 Report (New Reviewer)

Comments and Suggestions for Authors

The authors have revised the manuscript according to the reviewer's comments. Thus, the manuscript can be accepted after proofreading. 

Comments on the Quality of English Language

Need proofreading. 

This manuscript is a resubmission of an earlier submission. The following is a list of the peer review reports and author responses from that submission.

Round 1

Reviewer 1 Report

Comments and Suggestions for Authors

The manuscript "Japanese University Students are Highly Conscious of Infection Prevention Even After the Spread of COVID-19" reported a survey study to investigate the association between college students' vaccination coverage of the Omicron strain and their image of the vaccine and 68 past adverse reactions to the vaccine and to compare the results with those in 2021. Overall, the design of the study is average. The presentation of the result needs significant revision. Discussion should be closely related to this study, and the conclusion should be precise. Here are my comments for your reference.

1. Result 3.1, I would suggest changing " sequelae" to "symptom" as the variable listed is considered as "symptoms."

2. Result 3.2, the numbers and percentages in the text are different than in Table 2. It is very confusing. Please make sure the data in the table is the data you described in text.

3. Result 3.3: there is no figure legend to define what the color block means. I also can not find enough information to understand why the"with and without" were labeled on both the bottom and right sides when only presenting side effects and vaccinization. 

4. Result 3.4, comparison with 2021 data, is one of the core results of the study. It would be much easier for the reader to understand the data if data, including numbers, percentages, and p-values, were presented in the table while you can describe them in the text. 

5. The discussion is too long and should be shortened to the content closely related to this study.

6. Conclusion is to state your finding only, please do not include "guessing" or "hypothesis", such as "The fact that young adults who perform a wide 278 range of activities are careful about infection prevention may be one of the factors that 279 prevents the explosive spread of infection in Japan" There are no results support this statement.

Thanks

Jie

Comments on the Quality of English Language

English is understandable. It may be helpful for a few more proofreading.

Author Response

Dear Reviewer,

Thank you for your insightful comments on our manuscript. We feel the comments have helped us significantly improve the manuscript. 

Point 1: Result 3.1, I would suggest changing " sequelae" to "symptom" as the variable listed is considered as "symptoms."

Response 1: Indeed, as you pointed out, we changed “sequelae” to “sequelae symptoms” or “symptoms” in Result 3.1. (Line 92, 96,97, 100 indicated in red)

Point 2: Result 3.2, the numbers and percentages in the text are different than in Table 2. It is very confusing. Please make sure the data in the table is the data you described in text. 

Response 2: As you pointed out, the numbers in the text and the table were different. The numbers in the table were incorrect. We have corrected the error and added the information. (Table 2)

Point 3: Result 3.3: there is no figure legend to define what the color block means. I also can not find enough information to understand why the"with and without" were labeled on both the bottom and right sides when only presenting side effects and vaccinization. 

Response 3: We did not provide a legend and it was difficult to understand. We have provided a legend and corrected the axis labels on the graph. (Figure 3, 4)

Point 4: Result 3.4, comparison with 2021 data, is one of the core results of the study. It would be much easier for the reader to understand the data if data, including numbers, percentages, and p-values, were presented in the table while you can describe them in the text.

Response 4: Thank you for your valuable advice. I have added a comparison table with 2021 as advised. (Table 3)

Point 5: The discussion is too long and should be shortened to the content closely related to this study.

Response 5: As you pointed out, the discussion was redundant. We have tried to be concise by deleting unnecessary parts.

Point 6: Conclusion is to state your finding only, please do not include "guessing" or "hypothesis", such as "The fact that young adults who perform a wide 278 range of activities are careful about infection prevention may be one of the factors that 279 prevents the explosive spread of infection in Japan" There are no results support this statement.

Response 6: As you indicated, we have described only the findings in the conclusion. (Line 229-231)

Reviewer 2 Report

Comments and Suggestions for Authors

The paper is very interesting. I have few questions for authors and think that there are issues that must be clarified.

1.      One of the study aims was to compare the results of this cross-sectional study with that conducted in 2021. If it is the same settings and same methodology how it comes that in the year 2021were enrolled 2060 students and in 2023 only 1083. That is almost double difference. It also should be discussed in the study limitations.

2.      In the method section should be explained how many students were sent email, and how many of them were answered to survey. It was reported in method section that all the students at Hiroshima University were sent an e-mail, but we do not know how many are at the university?

3.      Advantages of the study (lines 75-78) should be moved to the end of the discussion, before the limitations. However, in the method section can be mentioned something about the study connected in 2021.

4.      In the statistical analysis there was not mentioned which statistical tests were used. Please, provide it.

5.      Authors should think to put one more table or graph for results presented in lines 166-175. These results are one of the study aims.

6.      Line 152 "according to the responses of 276 respondents who were vaccinated on campus". Does it mean that the responses were got only from these 276 respondents? How authors got these answers? Probably from students in the campus during vaccination? If it is so then methodology must be explained in much more details.

7.      Lines 85-86 "). When we sent the survey by personal 85 e-mail and requested their cooperation", it is not clear to which there is referred to? Probably students’ cooperation?

8.      In the beginning of the discussion section first should be mentioned the main results of this study and then compared with other studies. Not first to be presented results of the other studies.

Author Response

Dear Reviewer,

Thank you for your insightful comments on our manuscript. We feel the comments have helped us significantly improve the manuscript. 

Point 1: One of the study aims was to compare the results of this cross-sectional study with that conducted in 2021. If it is the same settings and same methodology how it comes that in the year 2021were enrolled 2060 students and in 2023 only 1083. That is almost double difference. It also should be discussed in the study limitations.

Response 1: Indeed, as you pointed out, there were 1083 fewer respondents in 2023 than in 2021. We have described this as a limitation of the study. (Line 221-224 indicated in red)

Point 2: In the method section should be explained how many students were sent email, and how many of them were answered to survey. It was reported in method section that all the students at Hiroshima University were sent an e-mail, but we do not know how many are at the university? 

Response 2: As you indicated, we did not state how many students we sent the e-mail to. We have added it to the Methods section. (Line 47)

Point 3: Advantages of the study (lines 75-78) should be moved to the end of the discussion, before the limitations. However, in the method section can be mentioned something about the study connected in 2021.

Response 3: As you indicated, we have moved the advantages of this study to the end of the discussion, before the limitations. (Line 218-220)

Point 4: In the statistical analysis there was not mentioned which statistical tests were used. Please, provide it.

Response 4: Sorry, the description of statistical methods was omitted, we have added. (Line 85-87)

Point 5: Authors should think to put one more table or graph for results presented in lines 166-175. These results are one of the study aims.

Response 5: Thank you for your valuable suggestions. We added Table 6 .

Point 6: Line 152 "according to the responses of 276 respondents who were vaccinated on campus". Does it mean that the responses were got only from these 276 respondents? How authors got these answers? Probably from students in the campus during vaccination? If it is so then methodology must be explained in much more details.

Response 6: Thank you for pointing this out. The answer to the question was to those that were vaccinated as a group on campus, so 276 people responded to the question. We have added an explanation in the text. (Line 122, 123)

Point 7: Lines 85-86 "). When we sent the survey by personal 85 e-mail and requested their cooperation", it is not clear to which there is referred to? Probably students’ cooperation? 

Response 7: As you indicated, I have corrected “their cooperation”, to “students’ cooperation”. (Line 59)

Point 8: In the beginning of the discussion section first should be mentioned the main results of this study and then compared with other studies. Not first to be presented results of the other studies. 

Response 8: Thank you for your valuable suggestions. We discribed the main results of this study at the beginning of the Discussion section, followed by a description of other studies.

Reviewer 3 Report

Comments and Suggestions for Authors

This was indeed a very novel and commendable study .I appreciate you taking the effort to study the behavioral patterns amongst students especially with regards to vaccinations and respect towards infection control measures ,especially now since the WHO announced the official end to the COVID 19 pandemic . 

I was just wondering if you found any thoughts about vaccination fatigue amongst students and were they tired of being mortally fearful of the virus or did they still believe about the fatality associated with this virus to themselves or to their older family members ?

Any particular thoughts about the type of vaccine to be taken such as mRNA vaccines vs non mRNA Vaccines and their side effect profile and effectiveness.

Thanks 

Author Response

Dear Reviewer,

Thank you for your warm comments. It will be an encouragement to us in the future. We will respond to your comments as follows.

Point 1: I was just wondering if you found any thoughts about vaccination fatigue amongst students and were they tired of being mortally fearful of the virus or did they still believe about the fatality associated with this virus to themselves or to their older family members ?

Response 1: Thank you for your valuable suggestions. I have added that the students were indeed tired of vaccines. (Line 204, 205). As for the fear of viral death, we thought it was a little different from being disgusted with fear, since it is not strong at the point where the number of severe cases is decreasing.

Point 2: Any particular thoughts about the type of vaccine to be taken such as mRNA vaccines vs non mRNA Vaccines and their side effect profile and effectiveness.  

Response 2: We understand that your question is very valuable and important. However, we do not have a precise answer to your question. We are sorry. We will leave it for the future.